# Impacts of Assimilating CYGNSS Satellite Ocean-Surface Wind on Prediction of Landfalling Hurricanes with the HWRF Model

Zhaoxia Pu [1,*], Ying Wang [1], Xin Li [1], Christopher Ruf [2], Li Bi [3] and Avichal Mehra [3]

1  Department of Atmospheric Sciences, University of Utah, Salt Lake City, UT 84112, USA; tulip.wang@utah.edu (Y.W.); xin.li@utah.edu (X.L.)
2  Department of Climate and Space Science and Engineering, University of Michigan, Ann Arbor, MI 48109, USA; cruf@umich.edu
3  NCEP Environmental Modeling Center, College Park, MD 20740, USA; li.bi@noaa.gov (L.B.); avichal.mehra@noaa.gov (A.M.)
*  Correspondence: zhaoxia.pu@utah.edu

**Abstract:** This study examines the impacts of assimilating ocean-surface winds derived from the NASA Cyclone Global Navigation Satellite System (CYGNSS) on improving the short-range numerical simulations and forecasts of landfalling hurricanes using the NCEP operational Hurricane Weather Research and Forecasting (HWRF) model. A series of data assimilation experiments are performed using HWRF and a Gridpoint Statistical Interpolation (GSI)-based hybrid 3-dimensional ensemble-variational (3DEnVar) data assimilation system. The influence of CYGNSS data on hurricane forecasts is compared with that of Advanced Scatterometer (ASCAT) wind products that have already been assimilated into the HWRF forecast system in a series of assimilation experiments. The effects of different versions of CYGNSS data (V2.1 vs. V3.0) on hurricane forecasts are evaluated. The results indicate that CYGNSS ocean-surface wind can lead to improved numerical simulations and forecasts of hurricane track and intensity, asymmetric wind structure, and precipitation. The impacts of CYGNSS on hurricane forecasts are comparable and complementary to the operational use of ASCAT satellite data products. The dependence of the relative impacts of different versions of CYGNSS data on optimal thinning distances is evident.

**Keywords:** data assimilation; hurricane; numerical weather prediction; CYGNSS; ASCAT

## 1. Introduction

Landfalling hurricanes cause significant damage and casualties in coastal and inland areas. Their intensity, track, and associated precipitation are the main focuses in weather forecasting. However, hurricanes usually originate and evolve over the ocean before they approach the coastline and make landfall, and sparse observations over the ocean commonly make hurricane forecasts very challenging. In recent decades, with the rapid development of observation systems, more data sources (e.g., in situ observations, satellites, and ground-based and airborne radars) have become available for monitoring and analyzing hurricanes (e.g., [1–3]). Among these data sources, satellite-measured ocean-surface winds—such as those from the previous NASA Quick Scatterometer (QuickSCAT) and the recent Advanced Scatterometer (ASCAT) from the European Organization for the Exploitation of Meteorological Satellites (EUMETSAT)—have advantages in observing hurricane evolution over the ocean, as they provide hurricane circulation data near the surface and also carry the signature of air–sea interaction. Studies have indicated that satellite-derived ocean-surface winds can have positive impacts on the track, intensity, and structure forecasts of hurricanes, including landfalling hurricanes, when they are assimilated into global and regional Numerical Weather prediction (NWP) models (e.g., [4–7]). Because of the significant influence of satellite-derived ocean-surface winds on weather forecasts, they have been included in the routine data that are assimilated in operational NWP practice.

New technologies continue to be developed to obtain more accurate ocean-surface wind products for better observations and predictions of hurricanes.

Specifically, ASCAT is a fan-beam aperture C-band radar scatterometer, providing two independent 550-km-wide swaths of backscatter retrievals. The ASCAT satellite-retrieved ocean-surface wind products from the EUMETSAT are provided through the Royal Netherlands Meteorological Institute (KNMI) with a horizontal resolution of 25 km (e.g., [8]). The data have been assimilated into the European Centre for Medium-Range Weather Forecasts (ECMWF) Integrated Forecasting System (IFS) since 2007, and have proven to be beneficial in tropical regions. To further improve the impacts of ASCAT data in the ECMWF system, research has been conducted on quality control, thinning schemes, observations/background error estimation, and assimilation algorithms (e.g., [9–12]). In particular, Lin et al. [12] assimilated ASCAT wind data with a resolution of 25 km in a global model with a thinning distance of 100 km, to reduce errors in spatially correlated error representativeness. When assimilated into the NCEP operational Gridpoint Statistical Interpolation (GSI) data assimilation system with the same thinning distance, ASCAT wind products also significantly improved wind and temperature forecasts [13]. Because of their benefit to the operational forecasts, ocean-surface wind data from the ASCAT have become a conventional data source for the NCEP Automated Data Processing (ADP) database, and have been assimilated in NCEP operations since 2007.

Scientists have continued exploring better measurements of wind speed over oceans with a research satellite mission. An outstanding example is the recent NASA Cyclone Global Navigation Satellite System (CYGNSS) mission. In contrast to scatterometry, which has a radar instrument aboard a satellite which sends a signal to the ground and measures the signal strength reflected back to it, CYGNSS consists of eight micro-satellites in a common circular low earth-orbit to receive both direct and reflected signals from Global Positioning System (GPS) satellites. The direct signals pinpoint CYGNSS observatory positions, while the reflected signals respond to ocean-surface roughness, from which wind speed is retrieved. With new technologies, CYGNSS has the ability to observe winds over hurricane inner-core regions and can detect high winds, thus allowing scientists to better investigate and predict hurricanes [14]. Various research based on CYGNSS data has been conducted [15–17]. Early studies showed that assimilating CYGNSS Level II satellite-retrieved ocean-surface winds into the regional NWP model, such as the NCEP Hurricane Weather Research and Forecasting (HWRF) regional model, has good potential to improve hurricane track and intensity forecasts [18–20].

As a natural extension of these previous studies, it is important to further evaluate whether the ocean-surface winds derived from CYGNSS (as a research satellite) could be a valid data source for potential operational NWP applications. In particular, as most previous studies emphasize the impact of the assimilation of CYGNSS data on analyses and forecasts of hurricanes over the ocean, we intend to further examine the impact of the assimilation of CYGNSS ocean-surface winds on the prediction of landfalling hurricanes with a regional operational model.

Specifically, using a recent version of the NCEP HWRF model (Version H220), our research focuses on (1) comparing forecast impacts between ASCAT and CYGNSS wind data when both data types are assimilated into HWRF in multiple hurricane cases, and (2) examining the relative impacts and dependent factors of assimilating the most recent CYGNSS data at different versions (e.g., versions 2.1 and 3.0) on the HWRF analyses and forecasts of landfalling hurricanes. A series of assimilation experiments are conducted with the GSI (Gridpoint Statistical Interpolation)-based three-dimensional hybrid ensemble-variational data assimilation (hybrid 3DEnVar, [21]) method to assess satellite-derived ocean-surface wind assimilation in HWRF. Four major landfalling hurricanes, Florence and Michael in 2018, and Laura and Delta in 2020, are used in this study. The model, data, and experimental design are described in Section 2. The results and discussion are presented in Sections 3 and 4. A summary and some remarks are offered in Section 4.

## 2. HWRF Model, Data, Experimental Design, and Data Thinning

### 2.1. HWRF Model and Data Assimilation System

The recent HWRF operational system version (H220), which was updated in the fall of 2020 at the NCEP Environmental Modeling Center, was utilized to conduct data assimilation and numerical simulations. The HWRF system was composed of the WRF (Weather Research and Forecasting) non-hydrostatic mesoscale model on an E grid dynamic core, the Message Passing Interface Princeton Ocean Model for Tropical Cyclones (MPIPOM-TC), and the NCEP data assimilation platform based on the Gridpoint Statistical Interpolation (GSI) system, with implementation of operational hybrid 3DEnVar capability. The physical parameterizations employed in the hurricane simulations were the Ferrier (new Eta) microphysics scheme, the Rapid Radiative Transfer Model for GCMs (RRTMG) longwave and shortwave radiation schemes, the Unified Noah land-surface model, the hybrid Eddy Diffusivity Mass-Flux (EDMF) Global Forecast System (GFS) scheme, and the scale-aware GFS Simplified Arakawa–Schubert (SASAS) convection scheme. The HWRF model and data assimilation system were executed in a cycling mode with a 6 h interval, producing forecasts of hurricane track, intensity, structure, and precipitation to meet operational forecast and warning process objectives. All details about the HWRF model and data assimilation system can be found in Biswas et al. [22]).

NCEP GSI-based hybrid 3-dimensional ensemble-variational (3DEnVar) data assimilation was employed as part of HWRF model initialization. The background error covariance of hybrid 3DEnVar was composed of a flow-dependent background error covariance achieved by 80 ensemble members from the Global Forecast System (GFS), and a static background error covariance obtained through the National Meteorological Center (NMC) method. The cost function is as follows:

$$J(x_1', \alpha) = \beta_1 (x_1')^T B_1^{-1} (x_1') + \beta_2 (\alpha)^T A^{-1} (\alpha) + \left( y^{0'} - Hx' \right)^T R^{-1} \left( y^{0'} - Hx' \right) + J_c, \quad (1)$$

where $B_1$ is the static background error covariance matrix; $\beta_1$ and $\beta_2$ are, respectively, the weight applied to the static background error covariance and the ensemble covariance; $\alpha$ contains the extended control variables for ensemble members; $A$ defines the spatial correlation of $\alpha$; $y$ is the innovation vector; $H$ is the observation operator; $R$ is the observational and representativeness error covariance matrix (with diagonal values for the observational errors); and $J_c$ is a constraint term. In HWRF operational system, $\beta_1$ and $\beta_2$ are set to 0.2 and 0.8, which offers more weight to the flow-dependent background error covariance.

### 2.2. Satellite-Derived Ocean-Surface Winds: CYGNSS vs. ASCAT

ASCAT MetOp-A/B operational level 2 vector wind products are available at a 25 km spatial resolution. The data products represent 10 m neutral stability winds and have been part of the conventional data for the NCEP Automated Data Processing (ADP) database. According to early evaluation (e.g., [20]), the Fully Developed Seas (FDS) wind speed products of CYGNSS are suitable for assimilating into a numerical weather prediction model for hurricane prediction. As with ASCAT, the near-surface wind speeds derived from the CYGNSS satellite were also referenced at a height of 10 m. The effective footprint size for CYGNSS is approximately 25 km × 25 km (e.g., [14]). Following the HWRF 6-hourly analysis-forecast cycles, both ASCAT and CYGNSS wind data were processed with an assimilation window ranging from −3.0 h to 3.0 h based on the HWRF analysis cycling window.

Figure 1 shows the ASCAT and CYGNSS data locations for Hurricanes Florence and Michael at 0000, 0600, and 1200 UTC on 13 September 2018, and at 0600, 1200, and 1800 UTC on 9 October 2018, respectively, when both hurricanes were near landfall. Both types of data were available over the ocean and near the coastline during most of the 6-hourly data assimilation windows, although they were unevenly distributed both spatially and temporally.

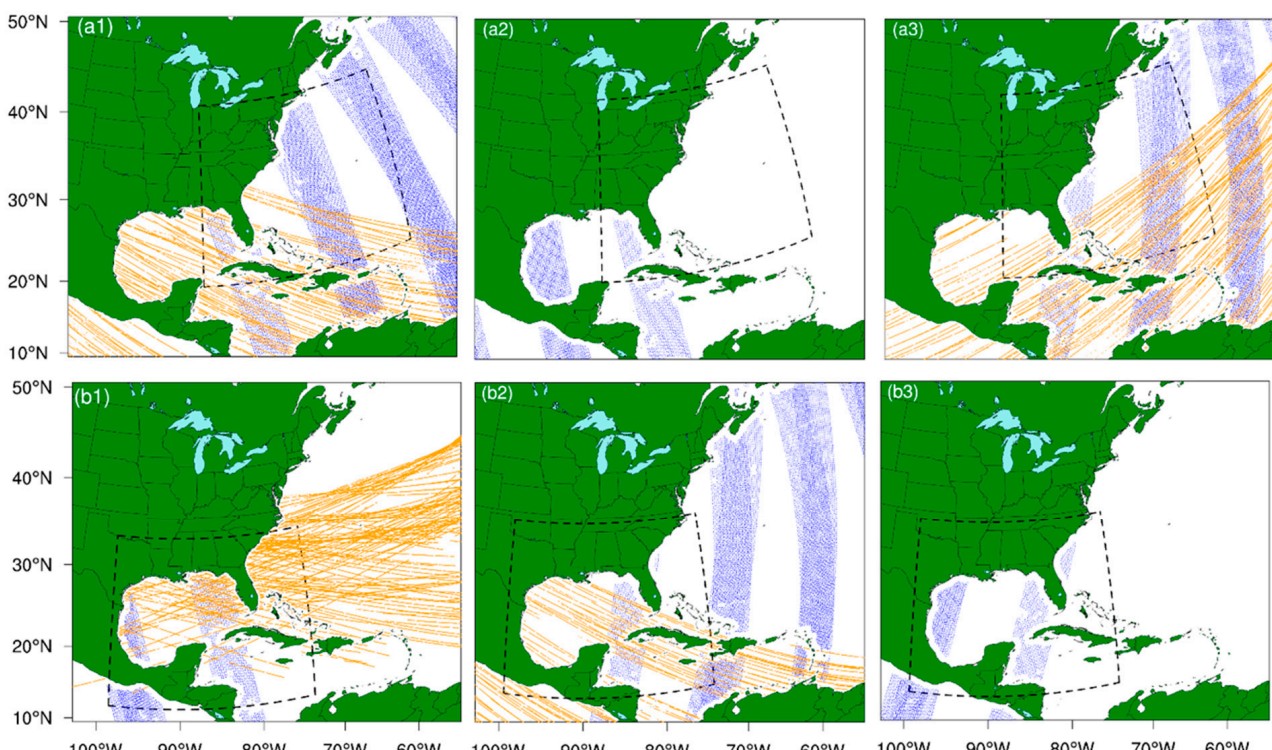

**Figure 1.** Available ASCAT and CYGNSS v2.1 data distribution for Florence and Michael at different assimilation times: (**a1**–**a3**) are, respectively, at 0000, 0600, and 1200 UTC on 13 September 2018; (**b1**–**b3**) are, respectively, at 0600, 1200, and 1800 UTC on 09 October 2018. Blue dots represent locations for ASCAT data and orange dots represent locations for CYGNSS data. The black dashed frames are the HWRF ghost domain for d02.

### 2.3. Data Thinning

When observations are assimilated into NWP models, observation errors are assumed to be uncorrelated, and each observation is an independent piece of information. Satellite observations may be oversampled and closer together than the product resolution. Assimilating such high-density information into NWP models will make the crucial assumption of uncorrelated observation errors invalid and lead to a degraded analysis [23–25]. Thus, thinning, or reducing the density of satellite observations, is usually applied in assimilation procedures. We introduced a thinning algorithm to the CYGNSS data when assimilating them into the GSI system. The resolution of both the CYGNSS and ASCAT ocean-surface wind data products was 25 km. However, unlike the two independent 550 km wide swaths by ASCAT, the oversampled observations achieved by the eight micro-satellites of CYGNSS within a 6 h assimilation time window were more complex (Figure 1). Therefore, it was necessary to test various thinning distances for CYGNSS ocean-surface wind data during the numerical experiments.

### 2.4. CYGNSS V2.1 vs. CYGNSS V3.0

With different ways to process and calibrate the CYGNSS Level I data, various versions of CYGNNS-retrieved ocean-surface data become available. For instance, following the CYGNSS V2.1 data, CYGNSS V3.0 data were available when we conducted this study, with a different calibration algorithm for CYGNSS Level I data (e.g., [26]). As revealed in Figure 2, the two versions (V2.1 and V3.0) of the CYGNSS FDS wind products show different error statistics: the V3.0 products include more high-wind data (>20 m·s$^{-1}$) but a smaller data volume compared to the V2.1 products. While the capability of representing high wind speeds is significantly improved in the V3.0 data products, large uncertainties are presented with these data as the wind speed errors are high. Because of these differences,

the forecast impacts with assimilating different versions of the CYGNSS data, and the thinning distances for these two versions, the data also needed to be tested in the numerical experiments (see Table 1).

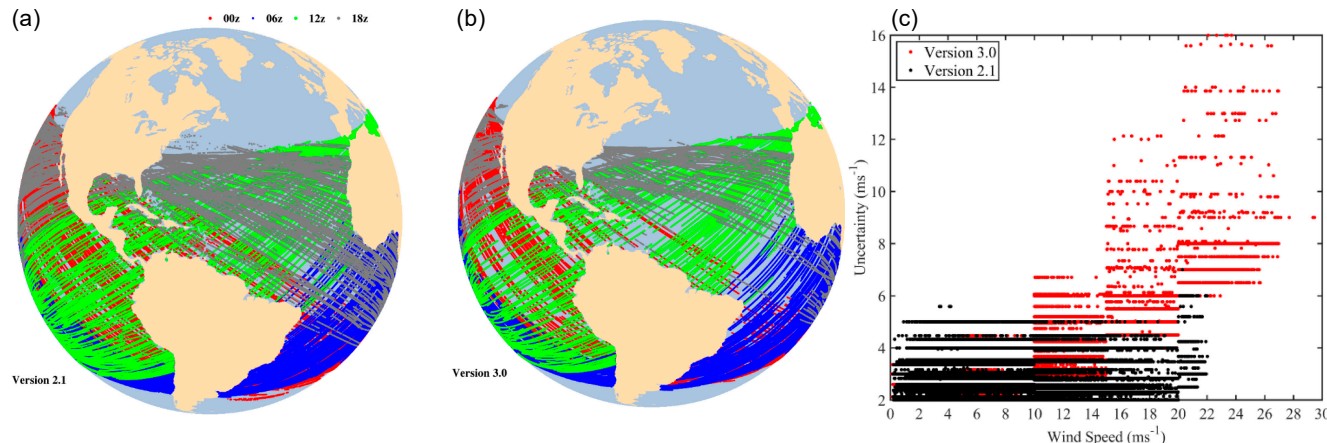

**Figure 2.** Available CYGNSS data of (**a**) version 2.1 and (**b**) version 3.0 on 13 September 2018 at different assimilation times, represented by different colored dots (red, blue, green, and grey colors for 00 UTC, 06 UTC, 12 UTC, and 18 UTC, respectively), and (**c**) comparison of uncertainty (the wind speed error) between these two versions of CYGNSS wind data (black and red colors denote version 2.1 and version 3.0 data, respectively).

**Table 1.** Configurations for all data assimilation experiments. ADP represents ADP conventional data used in NCEP operational analysis. In the experiment names, "F" stands for Florence, "M" for Michael, "L" for Laura, and "D" for Delta.

| Experiments | Data Assimilated | | | | CYGNSS Thinning Distance (km) |
| --- | --- | --- | --- | --- | --- |
| | ADP (Exclude ASCAT) | ASCAT | CYGNSS-v2.1 | CYGNSS-v3.0 | |
| Exp. FCOAS; Exp. MCOAS, Exp. LCOAS; Exp. DCOAS | X | X | | | - |
| Exp. FCOCY-2-50; Exp. MCOCY-2-50, Exp. DCOCY-2-50, Exp. LCOCY-2-50 | X | | X | | 50 |
| Exp. FCOASCY-2-50; Exp. MCOASCY-2-50, Exp. LCOASCY-2-50 | X | X | X | | 50 |
| Exp. LCOCY-2-30; Exp. FCOCY-2-30, Exp. LCOCY-2-30, Exp. DCOCY-2-30 | X | | X | | 30 |
| Exp. LCOCY-3-30; Exp. FCOCY-3-30, Exp. DCOCY-3-30 | X | | | X | 30 |
| Exp. LCOCY-3-50; Exp. FCOCY-3-50, Exp. DCOCY-3-50 | X | | | X | 50 |
| Exp. LCOASCY-3-30; DCOASCY-3-30 | X | X | | X | 30 |
| Exp. LCOASCY-3-50; DCOASCY-3-50 | X | X | | X | 50 |

Considering the discrepancies in their error characteristics, different observational errors were set for CYGNSS V2.1 and V3.0 data in the data assimilation system. Specifically, the observation error was set to 2 m·s$^{-1}$ for CYGNSS V2.1 data based on the statistics of the large data sample. For CYGNSS V3.0 data, the observation error was set to 3 m·s$^{-1}$ based on the statistics and sensitivity tests. Early experiments (not shown) found that the assimilation results were not sensitive to the varying observational errors of V3.0 data

because most of the data assimilated into the GSI system were still low winds in V3.0. Owing to the large errors, some high winds in V3.0 data were rejected by the quality control process inside the GSI system due to either significant errors of the data or large observations minus the background values.

*2.5. Experimental Design*

Four hurricane cases were used for the experiments. Florence and Michael represent the major landfalling hurricanes in the entire 2018 hurricane season, and Laura and Delta are the most notable landfalling hurricanes in the 2020 hurricane season. Florence formed on 31 August 2018, southeast of Santiago Island, in the southernmost Cape Verde islands. It moved to the North American continent with a western trajectory, landfalling along the southeastern coast of North Carolina near the upper end of category 1 [27]. Michael was a category 5 hurricane that made catastrophic landfall near Mexico Beach and Tyndall Air Force Base, Florida. The maximum sustained wind was about 140 kt at landfall time [28]. Hurricane Laura made landfall near Cameron, Louisiana, as a category 4 hurricane and continued its northward track into Arkansas before moving east toward the Atlantic. Hurricane Delta was a category 4 hurricane over the Atlantic on 6 October 2020, and it made landfall at 2300 UTC 9 October near Creole, Louisiana, with winds of 155 km h$^{-1}$ (~43 m·s$^{-1}$) and a pressure of 970 mb, as a category 2 hurricane.

The horizontal resolutions for HWRF's three simulation domains were 13.5 km, 4.5 km, and 1.5 km, respectively. The GSI-based hybrid 3DEnVar data assimilation system assimilated ocean-surface wind data in the second ghost domain (at a 4.5 horizontal km resolution), along with other observations assimilated in the operational domain. The landfall time for Florence was 1115 UTC on 14 September 2018; for Michael 1730 UTC on 10 October 2018; and for Laura 0700 UTC on 27 August 2020, and 2300 UTC on 9 October. In order to examine the impact of assimilating ocean-surface winds on the forecasts of landfalling hurricanes, all data assimilation experiments were initialized 36 h before landfall, after a day of spin-up with the regular HWRF analysis/forecast cycle. For Florence, cycled data assimilation experiments with ocean-surface winds were conducted at 0000, 0600, and 1200 UTC on 13 September 2018; for Michael at 0600, 1200, and 1800 UTC on 9 October 2018; for Laura at 1800 and 0000 UTC on 25 August and 0600 UTC on 26 August 2020; and for Delta at 1200 and 1800 UTC on 8 October and 0000 UTC on 9 October 2020. As in the HWRF operational configuration, a blending option was turned on. A vortex relocation was employed in the vortex initialization procedure before the data assimilation.

The initial sensitivity experiments were conducted with various data thinning distances (e.g., 25 km, 30 km, 50 km, and 100 km) in the data assimilation experiments for CYGNSS data. Compared with other experiments, results from the experiments with thinning distances of 30 km and 50 km seemed to be optimal in most cases. Therefore, for the experiments presented in this study, we used data thinning distances of 30 km and 50 km for CYGNSS data and further evaluated the sensitivity of assimilation results to the choice between the two distances. With different combinations of data types and thinning distances for CYGNSS, various experiments were designed for all four hurricanes (Table 1).

The relative impact of the CYGNSS version 2.1 and ASCAT satellite-retrieved ocean-surface wind products on hurricane analyses and forecasts were first evaluated. The operational data assimilation setting, which assimilates the ADP conventional data (with ASCAT ocean-surface winds, e.g., Exp. FCOAS, MCOAS, etc., in Table 1), was conducted and used as the baseline experiment. To compare the relative impacts of ASCAT and CYGNSS data on hurricane analysis and prediction, other experiments assimilated the ADP conventional data, with the ASCAST data being excluded and replaced by the CYGNSS ocean-surface wind data (e.g., Exp. FCOCY-2-50, FCOCY-2-30, etc., in Table 1). Additional experiments were performed to assimilate the ADP conventional data and CYGNSS ocean-surface wind data (namely, with both ASCAT and CYGNSS data, e.g., Exp. FCOACY-2-50, FCOCY-2-30, etc., in Table 1) into the HWRF model.

## 3. Impact of CYGNSS Ocean-Surface Winds on Analyses and Forecasts of Hurricanes

Based on the experiments listed in Table 1, especially the results from Hurricane Florence and Michael (2018), in this section, we first demonstrate the impacts of CYGNSS data on the analyses and forecasts of hurricanes and compare the results with those experiments assimilating the ASCAT data. Then, we emphasize the other results, especially those from Hurricane Laura and Delta (2020), to compare the data assimilation results from CYGNSS V2.1 and V3.0 data in next section. Specifically, considering their compatible spatial coverage of two types of data (i.e., CYGNSS V2.1 and ASCAT), we first focus the experiments with assimilation of V2.1 CYGNSS data and ASCAT data, to compare their relative and combined impacts on the analyses and forecasts of these landfalling hurricanes. To obtain a fair comparison for different cases regarding the data impact, the forecast evaluation emphasizes short-range (48 h) prediction.

### 3.1. Improved Hurricane Inner-Core Structure in Analyses and Forecasts

The asymmetric structure of the inner-core region plays an important role in the evolution, structure, and intensity change of hurricanes (e.g., [29]). With its advantage in measuring hurricane inner-core winds, we first evaluate the impact of assimilating CYGNSS on the representation of hurricane inner-core structure in the analyses and forecasts.

Figure 3 compares horizontal winds at the 17th eta level (approximately 925 hPa) from various experiments against the NOAA HRD P3 Radar horizontal wind field, at 500 m (Figure 3a), at 1200 UTC on 9 October 2018 for Hurricane Michael. The NOAA/ESRL High-Resolution Rapid Refresh (HRRR) operational analysis, at the same time, is also used as reference (Figure 3f) for comparison. HRRR analysis is the finest-resolution operational regional analysis available and provides reliable analysis for hurricane cases (e.g., [30,31]), as shown in Figure 3a,f. It is apparent that HRRR analysis agrees well with the NOAA P3 radar winds. An asymmetric wind field can be seen in both the P3 radar wind and HRRR analysis fields.

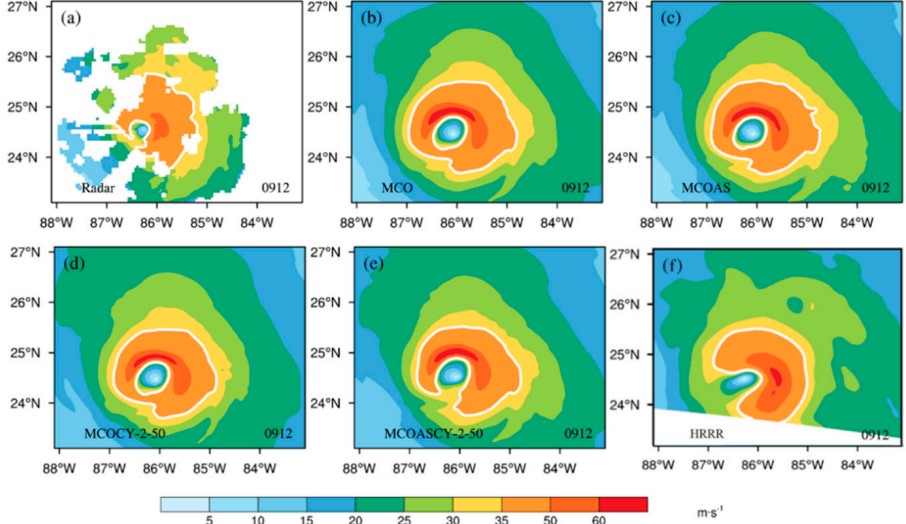

**Figure 3.** (**a**) NOAA P3 Radar horizontal wind field at 500 m and horizontal wind analysis field (unit: m·s$^{-1}$) at the 17th eta level (approximately 925 hPa) for Michael at 1200 UTC 9 October 2018. Panels (**b**–**e**) represent Exp. MCO, Exp. MCOAS, Exp. MCOCY-2-50, and Exp. MCOASCY-2-50, respectively. Panel (**f**) denotes HRRR analysis at 925 hPa.

To better compare the influence of the assimilation of ASCAT and CYGNSS wind data on hurricane structure, experiments (FCO for Florence and MCO for Michael) with the assimilation of ADP conventional data, excluding ASCAT data (no ocean-surface winds), are utilized for reference and comparison here. Without the assimilation of ocean-surface winds (Figure 3b), a relatively more symmetric structure (Figure 3b) is found in

the wind analysis. The assimilation of ASCAT does not lead to changes in this symmetric structure (Figure 3c), whereas the assimilation of CYGNSS data changes the inner-core structure toward asymmetry (Figure 3d), with the high winds (greater than 25 m·s$^{-1}$) more concentrated in the inner-core region. The asymmetric pattern is rebuilt in the southwestern portion of the vortex (Figure 3e) after the assimilation of both ASCAT and CYGNSS data. Although the maximum wind location is still displaced in the ASCAT and CYGNSS assimilation experiments (Figure 3c,d), assimilating CYGNSS ocean-surface winds along with ASCAT (Figure 3e) achieves reasonable asymmetric wind structure that agrees with the P3 radar wind field and HRRR analysis, especially in the southwestern portion of the vortex. A similar comparison is also made for Florence, Laura, and Delta (figures not shown), and most results indicate that the assimilation of CYGNSS ocean-surface winds, along with ASCAT data, leads to improved hurricane inner-core structure in the analysis field through cycled data assimilation.

Meanwhile, the asymmetric structure of hurricane winds at landfall is also important, as it is a significant factor for effective public warning. Figures 4 and 5, respectively, display the horizontal and vertical wind structure for Florence and Michael in HWRF forecasts at their landfall times. HRRR analysis is utilized as a reference for verification. In the HRRR horizontal wind field for Florence (Figure 4a), the maximum wind is in the western part of the core region. With the assimilation of all conventional (ADP) data without ASCAT or CYGNSS, Exp. FCO produces maximum wind in the core region not only in the western portion but also to the northeast. With the assimilation of ASCAT (Figure 4c), CYGNSS (Figure 4d), and both ASCAT and CYGNSS data (Figure 4e), the overestimation of winds in the core region is mitigated, implying more reasonable hurricane structure forecasts with the assimilation of ocean-surface winds. Similarly, for Hurricane Michael, the horizontal asymmetric wind structure is again missed in Exp. MCO, with an over-strong, tight, high-wind inner core. With ocean-surface wind assimilation in Exp. MCOAS, Exp. MCOCY-2-50, and Exp. MCOASCY-2-50, the strong wind in the inner-core region is significantly reduced (Figure 4h–j), especially in the southern and southwestern part of the core, where the winds are weak in the HRRR analysis.

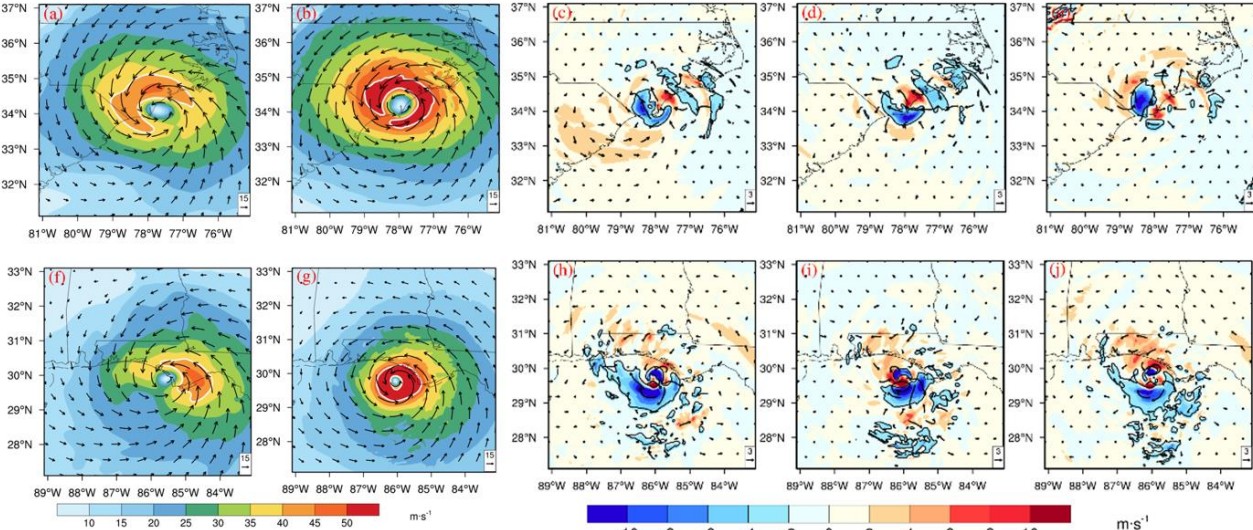

**Figure 4.** Horizontal wind (unit: m·s$^{-1}$) at 925 hPa for (**a**,**b**) Florence (at 1200 UTC on 14 September 2018) and (**e**,**f**) Michael (at 1800 UTC on 10 October 2018) at landfall. Colored contours represent wind speed, and vectors represent wind direction. Panels (**a**,**f**) the wind field of HRRR for Florence and Michael; panels (**b**,**g**) are the wind speeds and vectors for forecasts of Exp. FCO and Exp. MCO, respectively; panels (**c–e**) are the differences from Exp. FCO for the forecasts of Exp. FCOAS, Exp. FCOCY-2-50, and Exp. FCOASCY-2-50, respectively; and panels (**h–j**) are the differences from Exp. MCO for the forecasts of Exp. MCOAS, Exp. MCOCY-2-50, Exp. MCOASCY-2-50, respectively.

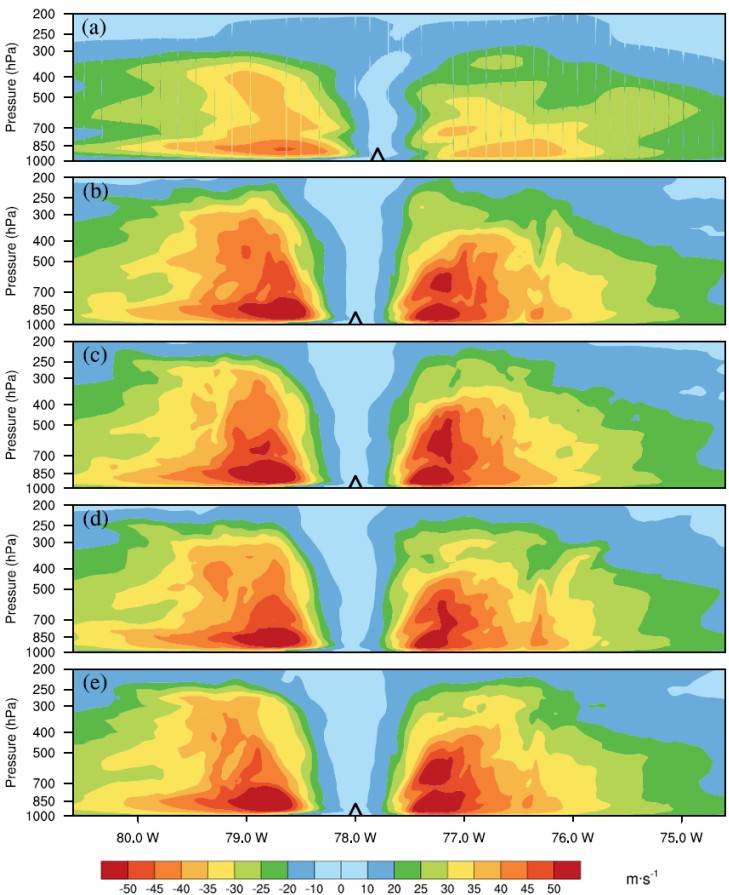

**Figure 5.** Wind field of vertical cross-section for Florence (at 1200 UTC on 14 September 2018) and Michael (at 1800 UTC on 10 October 2018) at landfall. Colored contours are wind speed (unit: m·s$^{-1}$). Panel (**a**) is the wind field of HRRR for Florence and Michael, and panels (**b**–**e**) are the forecast of Exp. FCO, Exp. FCOAS, Exp. FCOCY-2-50, and Exp. FCOASCY-2-50, respectively. The triangle represents the hurricane eye.

To examine the vertical structure of the hurricane vortex, we use a cross-section of wind fields for Florence as an example (Figure 5). The HRRR analysis shows that the strong wind extends higher on the western side of the vortex (Figure 5a), while the forecasts in the experiments without ocean-surface wind assimilation reproduce much stronger winds (Figure 5b). The simulations with the assimilation of ASCAT or CYGNSS ocean-surface winds reduce wind magnitude in the key areas of these core regions, thus enhancing the asymmetric structure of the hurricanes. Specifically, for Florence, with the assimilation of CYGNSS and ASCAT ocean-surface winds in Exp. FCOAS (Figure 5c), Exp. FCOCY-2-50 (Figure 5d), and Exp. FCOASCY-2-50 (Figure 5e), the vertical winds in the inner-core better represented the stronger winds near the surface and reduced the overestimated winds between 700–300 hPa relative to Exp. FCO. In addition, compared with the assimilation of CYGNSS data (Exp. FCOCY-2-50, Figure 5d), the influence on the winds from the assimilation of ASCAT data (Exp. FCOAS, Figure 5c) is relatively small on the east side of the inner-core. Similar results were observed with Hurricane Michael.

Overall, Figures 4 and 5 reveal the influences of assimilating ocean-surface winds from CYGNSS and ASCAT on the prediction of hurricane inner-core structure. Similar outcomes are also found in the other two hurricane cases (Laura and Delta, figures not shown). Cui et al. [20] found that the assimilation of CYGNSS data improves the representation of the hurricane inner-core asymmetric structure in forecasts. Although ASCAT data does not cover the hurricane inner-core region, assimilating the data into the hurricane environment seems beneficial to the vortex structure in the forecasts through model integration.

### 3.2. Impacts on Hurricane Track and Intensity Forecasts

Since ASCAT ocean-surface winds have already been assimilated into the HWRF system operationally, all other forecast experiments are compared against the experiment that assimilates ADP conventional data (including the ASCAT data; Exp. FCOAS and MCOAS). Figure 6 compares forecast tracks from data assimilation experiments for both Florence and Michael with the NHC best-track data. The short-range (48 h) track forecast is already nearly perfect, with the track errors at nearly 30 km in the experiment with the assimilation of all available conventional observations, including ASCAT data. Nevertheless, positive (about 10% to 20% track-error reduction) and neutral impacts on the track forecast are still found in most instances with the assimilation of CYGNSS data, compared with those experiments that assimilate ASCAT data. In particular, assimilating CYGNSS data with a thinning distance of 50 km (Exp. FCOCY-2-50 and Exp. MCOCY-2-50) performs better than assimilating ASCAT data (Exp. FCOAS and Exp. MCOAS; Figure 5a,b), as track errors are reduced by about 20% overall during the first 48 h forecast. Meanwhile, the assimilation of both ASCAT and CYGNSS data with other conventional data (Exp. FCOASCY-2-50) only results in about a 10% track-error reduction during the first 48 h forecast.

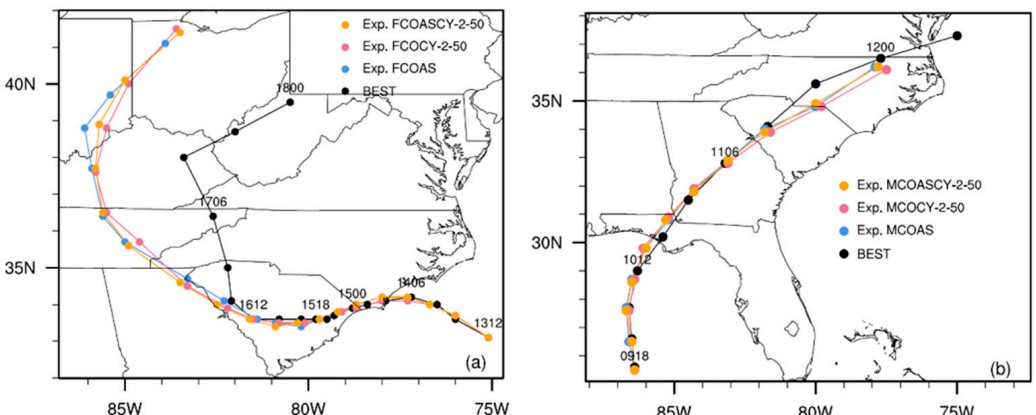

**Figure 6.** Forecast track of all assimilation experiments against the NHC best-track data (BEST) for Florence and Michael. (**a**) Florence (1200 UTC 13 to 0000 UTC on 18 September 2018); (**b**) Michael (1800 UTC 9 to 0000 UTC on 12 October 2018).

Figure 7 illustrates a time series of the forecast intensity (i.e., minimum hurricane center sea level pressure or MSLP, and maximum surface wind or MSW) for Florence and Michael. Positive and neutral impacts on the intensity forecast are found in most experiments with CYGNSS data assimilation compared to those experiments that assimilate the ASCAT data. Among all the experiments, MCOCY-2-50 for Hurricane Michael, with the assimilation of CYGNSS data using a thinning distance of 50 km, provides the best forecasts of minimum pressure and maximum wind at landfall. To obtain a more detailed comparison, Figure 8 shows the intensity (MSLP and MSW) errors averaged over the first 48 h forecasts. The assimilation of CYGNSS data outperforms the assimilation of ASCAT data in intensity forecasts. The improvements are small for Florence but rather large for Michael. When both ASCAT and CYGNSS data are assimilated into the HWRF model, the intensity forecast is still improved against the experiment that assimilates ASCAT data, but the positive impacts are slightly smaller compared to the assimilated CYGNSS data only.

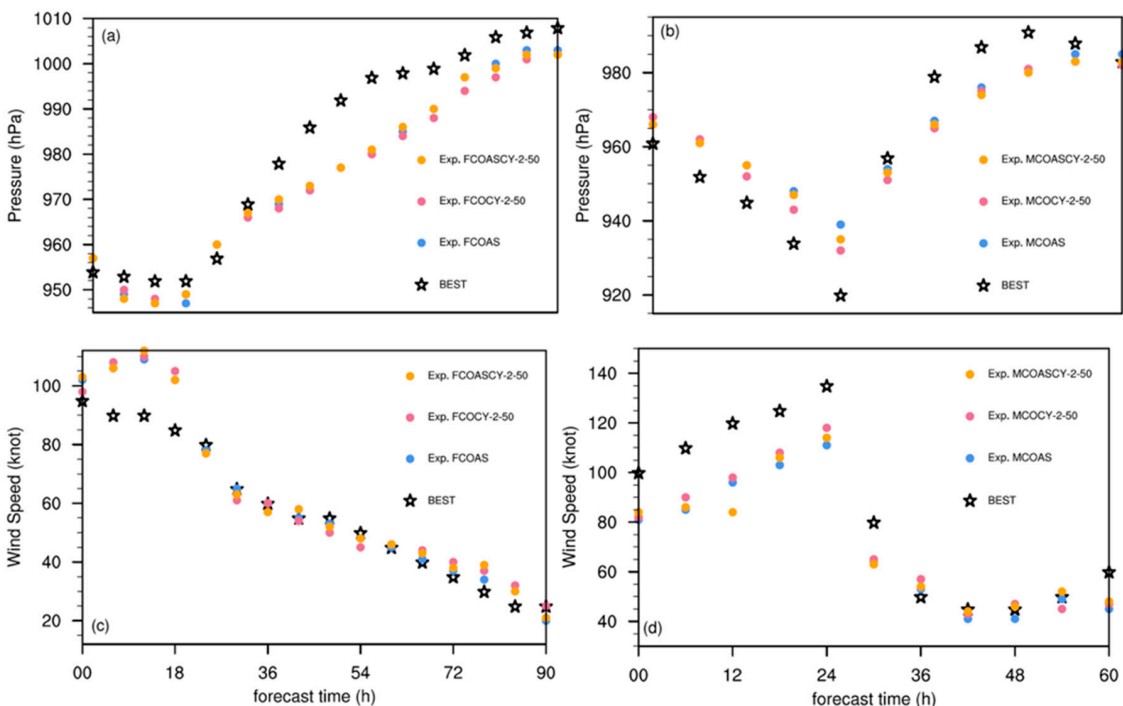

**Figure 7.** Forecast hurricane intensity for all experiments against the NHC best-track data: (**a**) the minimum hurricane center sea level pressure (MSLP; unit: hPa); (**c**) the maximum surface wind (MSW; unit: knot) for Florence (from 1200 UTC 13 to 0600 UTC on 17 September 2018); (**b**,**d**) are MSLP and MWS for Michael (from 1800 UTC 09 to 0600 UTC on 12 October 2018), respectively. *X* axis is forecast time (unit: h).

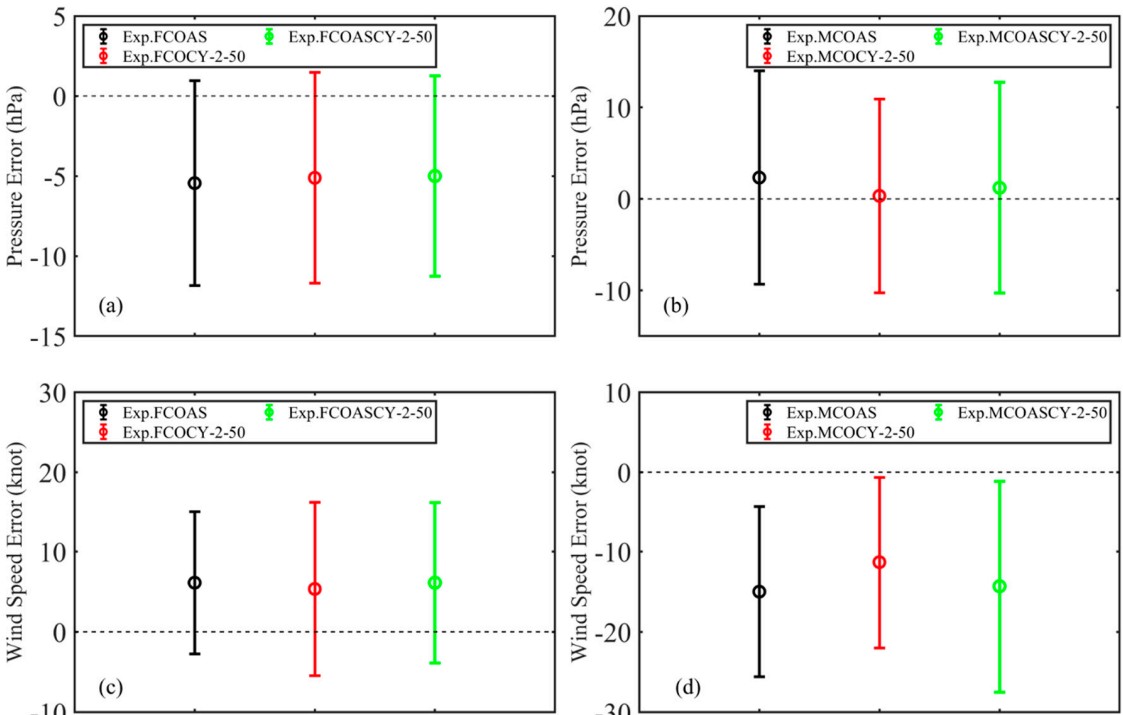

**Figure 8.** Average intensity errors, namely forecast against best-track data for (**a**,**b**) MSLP and (**c**,**d**) MSW over the first 48 h forecasts: (**a**,**c**) for Hurricane Florence (averaged from 1200 UTC 13 to 1200 UTC on 15 September 2018); (**b**,**d**) for Hurricane Michael (averaged from 1800 UTC 09 to 1800 UTC on 11 October 2018).

### 4. CYGNSS V2.1 vs. V3.0 Data

The results in the above section show that an appropriate thinning distance of 50 km for the CYNGSS V2.1 ocean-surface wind data leads to improved hurricane analyses and forecasts compared to ASCAT ocean-surface winds. With the CYGNSS mission development, different wind retrieval algorithms have been tested to produce the retrieved ocean-surface winds. As illustrated in Figure 2, the CYGNSS V3.0 wind product has increased high wind speed, but large uncertainty and less coverage. To compare the relative impact of the CYGNSS V3.0 and V2.1 data, HWRF analyses and forecast experiments (see Table 1) were performed for all hurricane cases used in this study, with various configurations for assimilating CYGNSS ocean-surface winds with all other data that are routinely assimilated with HWRF in an operational environment. As mentioned in Section 2.3, different thinning distances of 30 km and 50 km for these two versions of CYNGSS wind data are examined.

For Hurricane Florence, the simulated tracks (Figure 9a) from 1200 UTC on 13 September 2018 for Exps. FCOCY-2-30, FCOCY-2-50, FCOCY-3-30, and FCOCY-3-50 show that the track errors are small in the first 60 h. Exp. FCOCY-2-50 provides slightly better track forecasts, with the smallest track error in the first 48 h forecasts among all experiments. For the intensity forecasts (Figure 10), the assimilation of CYGNSS 2.1 or CYGNSS 3.0 data leads to notable improvements in the minimum sea level pressure forecast (Figure 10c) and a better forecast in the maximum surface winds (Figure 10d), compared with the experiment that assimilates ASCAT data. The assimilation of CYGNSS V3.0 data with a thinning distance of 30 km achieves the best intensity forecast among all the experiments, especially for the minimum sea level pressure forecast. The results indicate that a 50 km thinning distance seems to be optimal for the assimilation of CYGNSS V2.1 data, but a 30 km thinning distance is better for CYGNSS V3.0 data. Meanwhile, CYGNSS V3.0 data add more benefits to the intensity forecasts compared with the CYGNSS V2.1 data, because the V3.0 data lead to better wind and pressure forecasts in the inner-core region. As evidence, Figure 11 compares the HWRF forecasts of the wind profile with the NOAA P3 aircraft dropsonde data at 248 km from the hurricane center for Florence at 1200 UTC on 13 September. Apparently, the assimilation of CYGNSS data makes the simulated wind profiles closer to the dropsonde observations relative to the assimilation of ASCAT. Compared to the experiment assimilating the V2.1 CYGNSS data, the one that assimilates the V3.0 CYGNSS data produces inner-core wind profiles that are in better agreement with the dropsonde data.

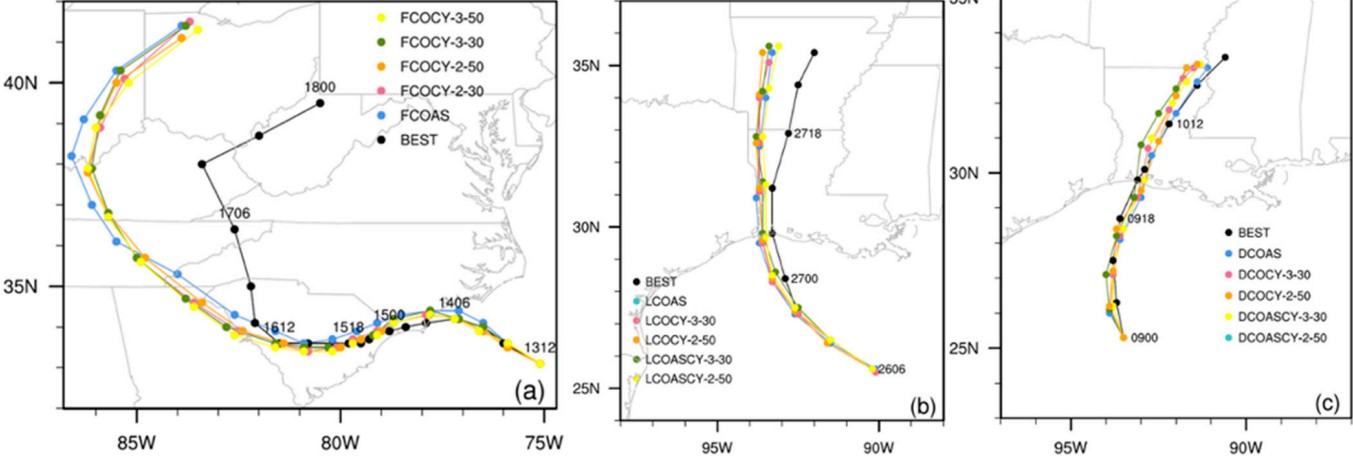

**Figure 9.** Same as Figure 6. Forecast track of all assimilation experiments against the NHC best-track data (BEST) for (**a**) Florence (1200 UTC 13 to 0000 UTC on 18 September 2018), (**b**) Laura (0600 UTC 26 to 0600 UTC on 28 August 2020), and (**c**) Delta (0000 UTC 09 to 0000 UTC on 11 October 2020).

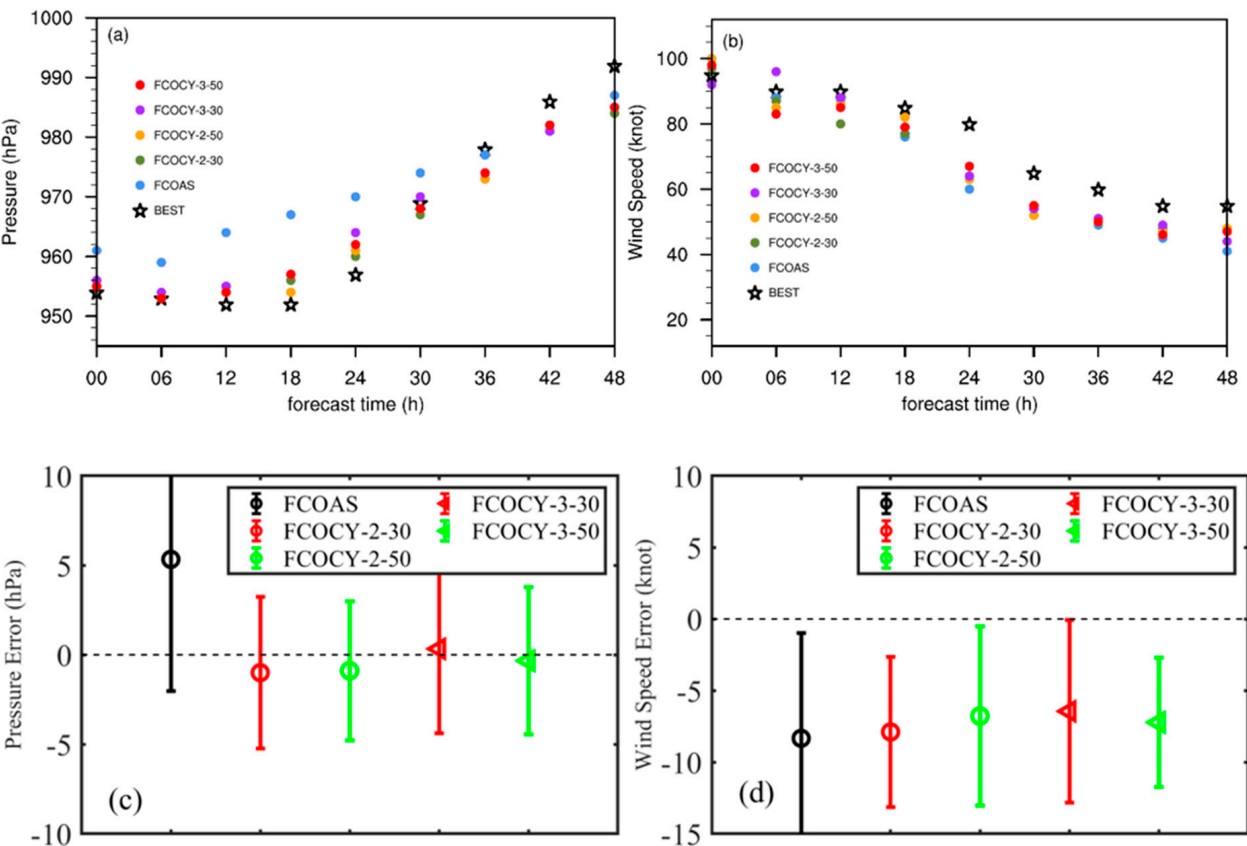

**Figure 10.** (**a**,**b**) Same as Figure 7, except for the time series of intensity for Florence from 1800 UTC on 9 to 1800 UTC on 11 October 2018; (**c**,**d**) Same as Figure 8, except for the 48-h average intensity errors for Florence. Figures show comparison and errors from different experiments (FCOCY-2-30, FCOCY-2-50, FCOCY-3-30, and FCOCY-3-50) against the NHC best-track data.

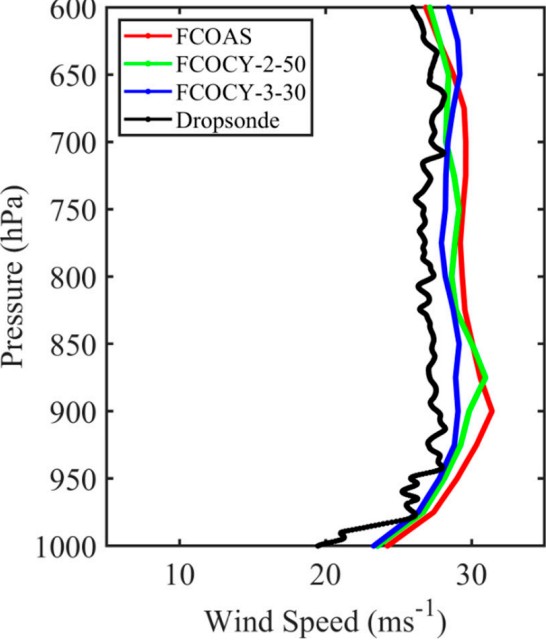

**Figure 11.** Wind profiles from HWRF forecasts compared with the NOAA P3 aircraft dropsonde at 248 km from the hurricane center for Hurricane Florence at 1200 UTC on 13 September.

Figure 12 reveals the 6-h accumulated rainfall from the different experiments against the Stage IV rainfall analysis. With the assimilation of ASCAT ocean-surface wind data, Exp. FCOAS misses the heavy rainfall southwest of the vortex and generates heavier rainfall in the northern part of the vortex (Figure 12b), compared with the Stage IV analysis (Figure 12a). With the CYGNSS data assimilation, both Exp. FCOCY-2-50 (Figure 10d) and Exp. FCOCY-3-30 (Figure 10e) have positive impacts on the rainfall forecasts. Exp. FCOCY-2-50 produces the best rainband precipitation among all the experiments by enhancing the rainbands in the eastern and southeastern part of the vortex and the southwestern part of the rainfall. Exp. FCOCY-3-30 produces more adjustment to the northern and part of the southern rainbands. Similar results to those for Florence are obtained for Hurricane Michael for the track, intensity, and precipitation forecasts (figures not shown) when both the CYGNSS V2.1 and V3.0 data are assimilated.

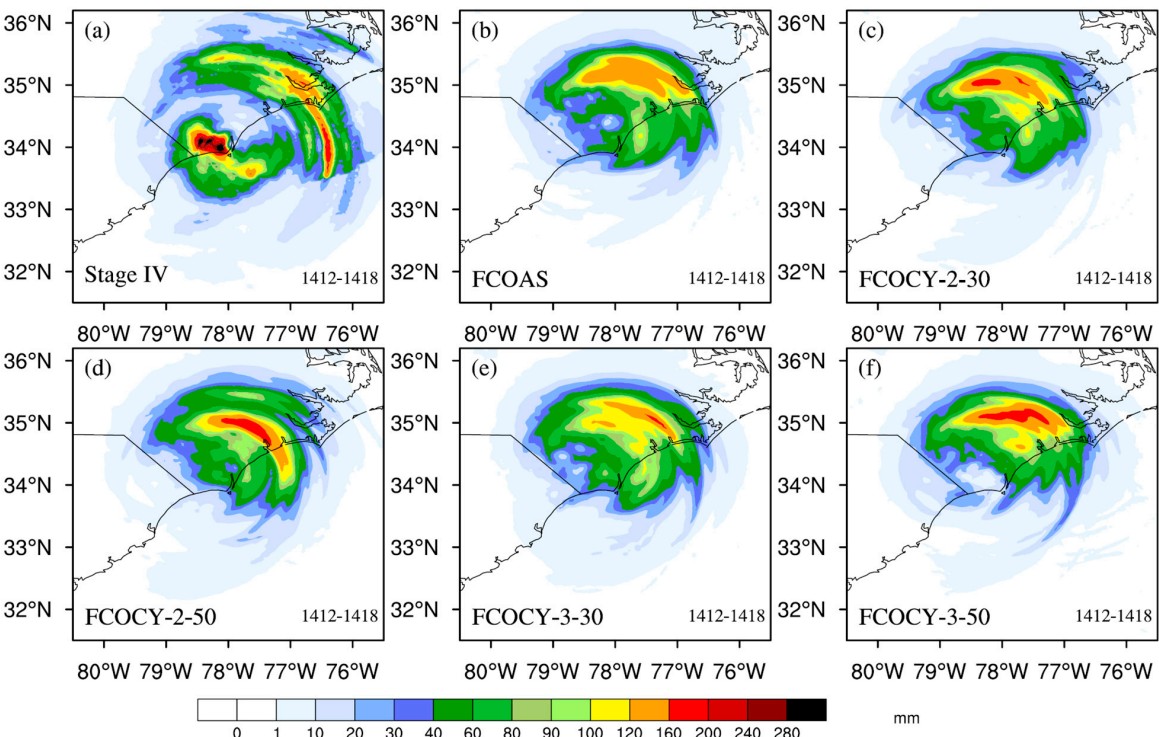

**Figure 12.** Six-hour accumulated precipitation (unit: mm) for Florence (from 1200 UTC to 1800 UTC on 14 September 2018) after landfall: (**a**) Stage IV2022 product; (**b–f**) forecast precipitation from experiments FCOAS, FCOCY-2-30, FCOCY-2-50, FCOCY-3-30, and FCOCY-3-50, respectively.

For Hurricane Laura, since thinning distances of 50 km for V2.1 CYGNSS data and 30 km for V3.0 data perform better than in the other experiments, the comparison focuses on these results. First, it is found that assimilation of ocean-surface winds from both ASCAT and CYGNSS leads to a comparable track forecast in the first 30 h forecasts (Figure 9b). Exp. LCOCY-2-50 produces the best track forecast among all the experiments with CYGNSS data assimilation, with a track-error reduction of about 10% in the first 18 h (out of a 50 km track error with LCOAS). Improvement in the intensity forecast (Figure 13) is also evident. Specifically, the assimilation of CYGNSS V2.1 data with a 50 km thinning distance leads to intensity-error reduction in both MSLP and MSW compared to the experiment that assimilates the ASCAT data. Combining both data sources in the assimilation experiment LCOASCY-2-50 also leads to a positive influence on the intensity forecast. However, the assimilation of CYGNSS V3.0 data with a 30 km thinning distance leads to a slightly degraded intensity forecast in this case. Combining CYGNSS V3.0 data with a 30 km thinning distance with ASCAT data in the data assimilation experiment mitigates these negative impacts on the intensity forecasts of Laura.

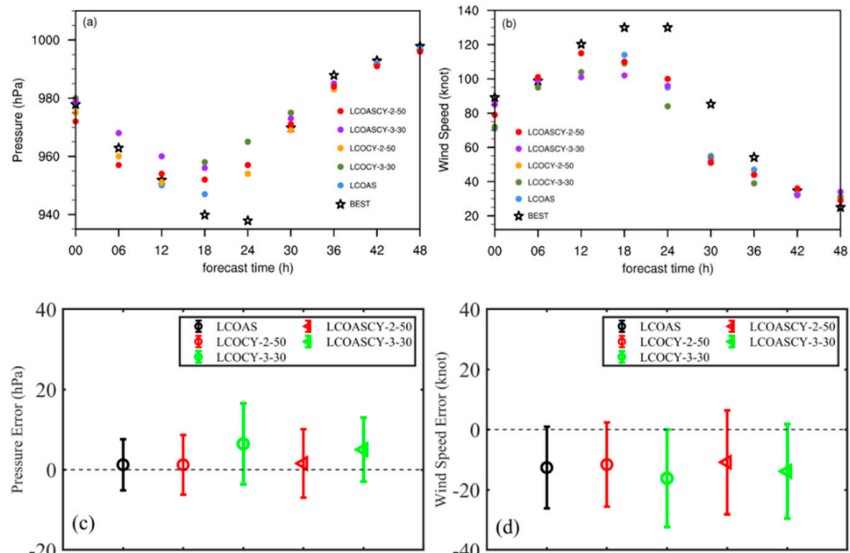

**Figure 13.** Same as Figure 10, except for intensity forecasts for Laura (0600 UTC on 26 August to 0600 UTC on 28 August 2020) from experiments LCOAS, LCOCY-2-50, LCOCY-3-30, LCOASCY-2-50, and LCOASCY-3-30 km, respectively.

For Hurricane Delta (2020), as shown in Figure 9c, the track errors are small (about 50 km) in all experiments within a 48 h forecast. The assimilation of ASCAT data in Exp. DCOAS and the CYGNSS V2.1 data with a 50 km thinning distance (Exp. DCOCY-2-50) leads to a very similar track forecast, while the assimilation of both sets of data in Exp. DCOASCY-2-50 leads to a 17% track-error reduction compared with both these experiments, during the first 48 h forecast. The assimilation of CYGNSS V3.0 data with a 30 km thinning distance (Exp. DCOCY-3-30) leads to about a 20% track-error reduction (10 km) on Delta's track forecasts, while Exp. DCOASCY-3-30 has a neutral impact on the track forecast. As for the intensity forecast (Figure 14), compared with Exp. DCOAS, DCOCY-2-50 and DCOCY-3-30 lead to slight positive impacts while the impacts from DCOCY-3-30 are more noticeable. Exp. DCOASCY-2-50, and DCOASCY-3-30 lead to slight positive or neutral impacts in the intensity forecast within the 48 h.

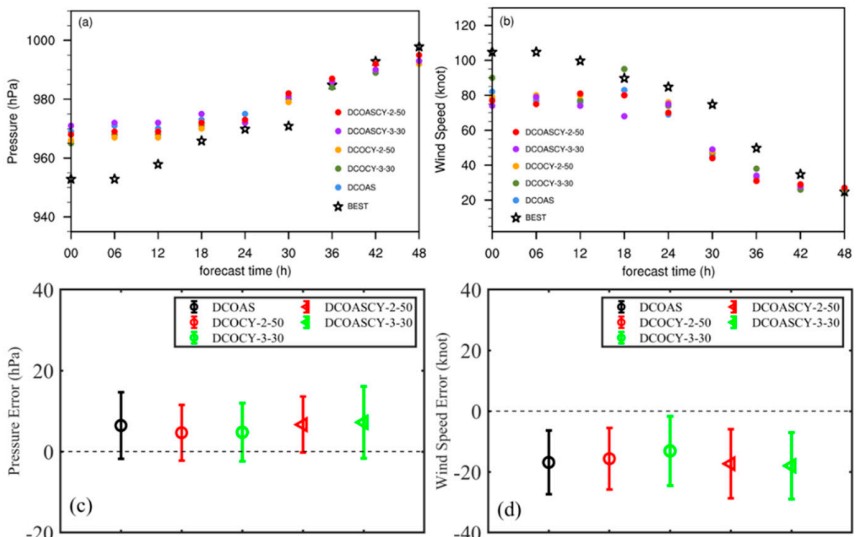

**Figure 14.** Same as Figure 10, except for intensity forecasts for Delta (0000 UTC 09 to 0000 UTC on 11 October 2020) from experiments DCOAS, DCOCY-2-30, DCOCY-2-50, DCOCY-3-30, and DCOCY-3-50, respectively.

## 5. Summary and Remarks

Leveraged by GPS satellites, the NASA research satellite CYGNSS, launched in 2016, has broadened the horizons of scientists investigating hurricanes with measurements of ocean-surface winds. In this study, the retrieved Level-II CYGNSS ocean-surface winds are assimilated into the NCEP regional hurricane HWRF forecast system to examine their impacts on predicting landfalling hurricanes. Four major hurricane cases, including Florence and Michael in 2018, and Laura and Delta in 2020, are used to conduct various assimilation experiments. Different versions of CYGNSS-retrieved ocean-surface winds are evaluated. Since ASCAT (on-board MetOp-A/B) provides real-time ocean-surface wind products for current operational forecasts, the impacts of CYGNSS data on hurricane prediction are compared with those of the ASCAT data.

With four major hurricane cases, it is found that the assimilation of CYGNSS Level-II ocean-surface winds has positive impacts on hurricane track, intensity, and precipitation forecasts. The assimilation of CYGNSS data also improves hurricane inner-core structure. Assimilating CYGNSS data provides forecasts of hurricane track comparable to those obtained by assimilating ASCAT data. Specifically, compared with the assimilation of ASCAT data, assimilating CYGNSS data can lead to improved track and intensity forecasts for a short range (within 48 h) when the proper thinning distances are applied. Because of the positive impacts of CYGNSS data in the analyses and forecasts, as well as its positive complementary nature to ASCAT data, ocean-surface wind product from CYGNSS shows potential to be employed as a conventional data source in the HWRF model. Meanwhile, there are also benefits to combine CYGNSS and ASCAT data together, and assimilate them into the HWRF system for better hurricane forecasts.

With the different ways to calibrate CYGNSS Level-I data in a research environment, the characteristics of the retrieved Level-II ocean-surface winds vary in data coverage, retention of high winds, and error ranges. Evaluation of the overall quality of different versions of CYGNSS data is beyond the scope of this study. However, a series of numerical experiments with multiple hurricane cases in this study indicates that both Versions 2.1 and 3.0 of CYGNSS-retrieved ocean-surface winds can have positive impacts on the predicted track, intensity, and precipitation of hurricanes in most cases. The dependence of the relative impacts of different versions of CYGNSS data on optimal thinning distance is evident. Specifically, an appropriate thinning distance for CYGNSS V2.1 wind data is 50 km, while that for the CYGNSS V3.0 wind data is 30 km with the current HWRF analysis and forecast cycles. The relative impacts of the different versions of CYGNSS data on hurricane forecasts are complicated. Assimilation of the V2.1 data with 50 km thinning shows the best improvement in hurricane track, intensity, and precipitation in most cases, reflecting better coverage and less uncertainty in the data (see Figure 2). Meanwhile, CYGNSS V3.0 data are beneficial to intensity in some cases, owing to the presence of higher wind data compared to the V2.1 data.

In summary, we evaluated the impacts of assimilating CYGNSS data on the analysis and prediction of landfalling hurricanes with four major hurricanes over Atlantic Ocean. The results indicate the potential to include CYGNSS ocean-surface wind in operational systems for hurricane forecast improvement, pending more comprehensive evaluation and proof of statistical significance in future work. With current and future developments in CYGNSS wind products (e.g., the availability of CYGNSS V3.1 data as this study is completed) and the HWRF regional hurricane forecast system, more studies and cases can be used to explore the best way to assimilate these data for improved hurricane analyses and simulations with HWRF and other operational models.

**Author Contributions:** Conceptualization, Z.P. and C.R.; methodology, Z.P., Y.W., X.L. and L.B.; software, Z.P., Y.W., X.L. and L.B.; validation, Y.W., X.L., L.B. and Z.P.; formal analysis, Y.W. and X.L.; investigation, X.L., Y.W. and Z.P.; resources, Z.P., C.R., L.B. and A.M.; data curation, Z.P. and C.R.; writing—original draft preparation, Z.P., Y.W. and X.L.; writing—review and editing, Z.P. and C.R., L.B. and A.M.; visualization, X.L. and Y.W.; supervision, Z.P.; project administration, Z.P.;

funding acquisition, Z.P. and C.R. All authors have read and agreed to the published version of the manuscript.

**Funding:** This study was supported by the NASA CYGNSS Science Team managed through the University of Michigan under NASA Award 80LARC21DA003 (Ruf and Pu) and NOAA Award # NA19OAR4590239 (Pu, Li, Bi, and Metha), and NSF Grant # OAC-2004658 (Pu and Wang).

**Data Availability Statement:** The NCEP ADP data were obtained from the website (https://rda.ucar.edu/datasets/ds337.0/, accessed on 26 February 2022). The CYGNSS data products are available from the NASA Jet Propulsion Laboratory's Physical Oceanography Distributed Active Archive Center at https://podaac.jpl.nasa.gov/CYGNSS (accessed on 26 February 2022).

**Acknowledgments:** Computational resources and support from the NOAA Jet supercomputer system (Pu), NCAR CISL Cheyenne supercomputer system, and the Center for High-Performance Computing (CHPC) at the University of Utah are appreciated.

**Conflicts of Interest:** The authors declare no conflict of interest.

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
