# Peer review of "Impacts of Assimilating CYGNSS Satellite Ocean-Surface Wind on Prediction of Landfalling Hurricanes with the HWRF Model"

_remotesensing, doi:10.3390/rs14092118_

Round 1
Reviewer 1 Report
This is s solid piece of scientific work with important potential application, namely to improve the NCEP hurricane forecasts. I have only a few clarifying questions that can probably be answered easily:
-
Line 140: “Following the HWRF 6-hourly analysis-forecast cycles, both ASCAT and CYGNSS wind data are processed with an assimilation window ranging from -3.0 h to 3.0 h based on the HWRF analysis cycling window.” This seems to suggest that all observations from a -3 to +3 hr time window are aggregated to a single wind field at analysis time prior to assimilating. Is that correct? If so, why is this done? And how can you assimilate observations in the future (+3hr) in an operational setting?
-
Line 185: “Specifically, the observation error is set to 2 ms-1 for CYGNSS V2.1 data based on the statistics of the large data sample. For CYGNSS V3.0 data, the observation error is set to 3 ms-1 based on the statistics and sensitivity tests.” Are these average errors? If so, how are these average errors used? According to Figure 2, the uncertainty for high winds are much larger. Would it be possible to use relative errors?
-
Line 236: “The initial sensitivity experiments were conducted with various data thinning distances (e.g., 25 km, 30 km, 50 km, and 100 km) in the data assimilation experiments for CYGNSS data.” How was the thinning done? Linear interpolation? Nearest neighbour?
Author Response
Please see attached response.

Reviewer 2 Report
This paper examines the impact of CYGNSS ocean surface winds on the prediction of landfalling hurricanes using the HWRF/GSI system. The experimental design and assessment of the overall impact has some merit. It is encouraging to see that CYGNSS ocean surface winds can provide comparable impact as the ASCAT ocean surface winds. However, I found that the results among various experiments are rather similar, or at best, marginal differences. The question is then, do most of the marginal differences really matter? Some of the conclusions are not evidently supported or overstated. Fundamentally, the paper aims to address the impact of CYGNSS ocean surface winds on landfalling hurricanes, does CYGNSS ocean surface wind impact landfalling hurricanes in different ways than hurricanes over the ocean?
A general courtesy to properly label the figures throughout this paper would help the readers to navigate various subplots. An example of Figure 4. A row title for Florence, with each panel labelling a) HRRR, b) control, c) ASCAT-control, d) CYGNSS-control, similarly for the second row for Michael.
Various experiment names in Table 1 are extremely confusing and really hard to follow. There is no need to label the Hurricane name in the experiment since throughout the discussion each storm is addressed separately. Suggest remove the ‘F’, ‘M’, ‘L’ and ‘D’ and list the experiment by storm name. For each storm, list exactly the experiments that were conducted. The common experiment names are (just suggestions here): NoASCAT, ASCAT, CYGNSS, BOTH. Then label the thinning and version for each common experiment same as what is currently used in the paper.
L293-294: Comparing Fig. 3c and Fig. 3d, there is visually no difference to me. Authors claim that “the assimilation of CYGNSS data changes the inner-core structure toward asymmetric (Fig. 3d), with the high winds more concentrated in the inner-core region” is visually not supported.
Figure 3a the wind field with no assimilation of ASCAT or CYGNSS still presents a decent asymmetry, with peak winds in the north of the vortex center. The assimilation of ASCAT, CYGNSS or both didn’t change much of the peak winds position or strength, rather modify the peripheral wind structure. Authors’ claim in L294-300 is evidently not supported in Figure 3. I would suggest a quantitative comparison, e.g. spectral analysis of the fields, to provide the asymmetry improvement that the authors claimed. This suggestion perhaps also goes to the precipitation structure in Fig. 12.
L304-L307: More importantly, why the combination of the CYGNSS and ASCAT would achieve a superior analysis (if it is true as authors claimed) than either CYGNSS or ASCAT alone? The authors explained it in two places L304 “CYGNSS data cover the hurricane inner-core region” and L354 “ASCAT data does not cover the hurricane inner-core region”. I believe operational HWRF v220 removes the increments from the inner core data, if it’s true, how does the CYGNSS data impact the wind structure in this case? An easy-to-follow explanation is needed here.
Fig.5: are wind vectors in (c-d) (g-h) the differences between the ASCAT/CYGNSS and control? If not, please clarify. From Figure 5, it is hard to see whether the assimilation of ASCAT/CYGNSS leads to more asymmetry of the wind structure. In fact, from Fig. 5g, the assimilation of ASCAT seems to bring more symmetry to the tangential wind structure (stronger wind on the east side of the core in Fig. 5f is reduced in Fig. 5g). In general, it is a good way to present the difference between control and new experiment, however, in this case, it is hard to visually tell an improved asymmetry. If wind vectors are not discussed at all, the full fields of tangential wind fields from each experiment should be presented, along with the differences to the control one.
In the response to reviewers, please provide the full fields of tangential wind speed from the experiments that assimilate ASCAT or CYGNSS in horizontal view in Figure 4 and in vertical view in Figure 5.
From discussion in Fig.3 to Figs. 4-5, the emphasis from Fig. 3 is that a) CYGNSS is capable of providing similar impact as ASCAT, and b) combing CYGNSS and ASCAT would achieve even better hurricane inner core structure. Is the latter point not evident in the wind structure presented in Fig. 4-5?
L366-367: “positive (about 10% to 20% track error reduction) and neutral impacts on the track forecast are still found in most cases with assimilation of CYGNSS data compared with these experiments assimilate ASCAT data.” Authors could easily reproduce the same quantitative comparison in Figure 8 for track forecasts.
L365-374: This part of the paragraph states the impact of CYGNSS on the track forecasts, only by providing percentage improvement relative to the control. To complete the full story, please provide a table to quantitatively list the mean and standard deviation of the forecast track absolute error during the first 48h, as well as percentage improvement/degradation, for all the cases/experiments.
L383-37: Similarly, the above suggestion for the table should include pressure and wind speed forecast errors and percentage improvement/degradation.
Fig.8: how many sample that goes into this averaging? Is it only 8 or 9 cycles? Please be specific in either the text or figure title. It is clear that none of the comparison here is statistically significant. Authors should state such limitation in the text.
L431-432: “FCOCY-2-30, FCOCY-2-50, FCOCY-3-30, and FCOCY-3-50 show that Exp. FCOCY-2-50 provides better track forecasts with the smallest track error in the first 48 h forecasts.” Figure 9 shows virtually very little difference among various experiments.
Fig. 10: Colors for various experiments in (a-b) should be consistent with (c-d).
L443-447: a dropsonde verification is good, but such verification at this specific distance seems a bit random. Where does the dropsonde come from? If it is from a reconnaissance mission, there are usually many dropsondes released in and around a hurricane center. Is this dropsonde assimilated in any of the experiment? i.e. is this an independent comparison? How does the verification work for other dropsondes? From the verification of just one random selected dropsonde to conclude that “V3.0 data lead to better wind and pressure forecasts in the inner-core region”, which seems to be an exaggerated statement. What does the difference of these wind profiles like compared to the dropsonde? What’s the mean root-mean-square error (RMSE)? What’s the average RMSE for all the dropsondes in this cycle?
L464-469: I would argue the assimilation of CGNSS actually overestimates the northern part of the rainband (Fig.12 c, d, f), compared to the ASCAT experiment (Fig. 12b). All in all, I don’t see much difference in Fig. 12 b-f. They all missed the heavy precipitation in the southwest of the hurricane center.
Reviewer 3 Report
The manuscript describes an attempt to assimilate ocean wind surface measurements from GNSSR observations provided by NASA's CYGNSS mission. A comparison of the impact of the assimilation of CYGNSS and ASCAT observations is provided. Although CYGNSS is not an operational mission, there are commercial missions in preparation that will soon deliver GNSSR for operations. In that respect, this study can be very useful and worth publishing.
We found, however, the experimental design overly complex. Since the objective is to compare the forecast impact of both CYGNSS and ASCAT data, the introduction of different version of CYGNSS data processing as well as varying the thinning distance is only adding confusion to the study. In a preliminary study, the authors should test and decide which version and thinning radius is best for CYGNSS observations. Once those two parameters are set, then the CYGNSS data can be used in the comparison with ASCAT.
The comparison is conducted over four Hurricane cases. While this is a relatively small sample, we encourage the authors to present statistical results, rather than enumerating the cases with positive/negative impact. There is no benefit for the reader to learn that CYGNSS data has a stronger impact on Hurricane Laura than on Hurricane Delta, unless a physical explanation is offered. If such reasoning is not possible, then a statistical approach across all the case is recommended.
We noted some variations in the writing style in the manuscript. Some sections need to be revised and rewritten. More specifically:
Line 19-23: The second paragraph of the Abstract is disconnected from the first paragraph and does not bring any new information. It can just be deleted.
Line 43: The acronym NWP has not be defined.
Line 48: "fan beam" is repeated in the same sentenced: "Specifically, ASCAT is a fan beam aperture C-band fan beam radar scatterometer,"
Line 49: "error" is missing at the end of "spatially corre- 58 lated representativeness."
Line 67-70: This phrase is too long, please break it apart
Line 73: "With new technologies", what are those technologies?
Line 77: "into the regional NWP model" which regional model is referred here?
Line 136-137: This sentence is very general and not really appropriate in this particular section
Line 152-154 and 167-169: The sentences are completely identical. Please remove this repetition.
Line 180-181: "large wind uncertainty has also been introduced." What is this uncertainty? please expand.
Line 192: Same as above, what is the uncertainty referred here?
Figure 2c: It's not clear what is plotted here.
